# Embodied Resistance: Multiracial Identity, Gender, and the Body

**Gabrielle G. Gonzales**

Department of Sociology, University of California Santa Barbara, Santa Barbara, CA 93106, USA;
gabrielleggonzales@ucsb.edu

**Abstract:** This article explores the importance of the physical body in the development of gendered racial and ethnic identities through in-depth semi-structured interviews with 11 multiracial/ multiethnic women. From a critical mixed race and critical feminist perspective, I argue that the development of an embodied and gendered multiracial and multiethnic identity is a path to questioning and resisting the dominant monoracial order in the United States. Interviews reveal that respondents develop these embodied identities both through understandings of themselves as gendered and raced subjects and through relationships with monoracial individuals. The process by which these women understand their physical bodies as multiracial subjects illustrates a critical embodied component of the social construction of race and ethnicity in the United States.

**Keywords:** multiracial; gender; feminist; embodiment; racialization; identity; resistance

---

## 1. Introduction

The scholarly research on and public discussion of mixed-race identity has increased in popularity within the last 50 years as the population and visibility of mixed-race individuals has also increased. Multiracial scholar Jackson (2012) states, "Due to the growing number of persons who claim membership in more than one racial group and the increased visibility of multiracial persons in the media in the late 1990s and early 2000s (e.g., Tiger Woods, President Barack Obama), multiracial identity has received increased attention in social science research" (p. 45). Although this is rapidly changing, the majority of research conducted on multiracial people is on biracial Black and white as well as white and Asian American individuals.

This article explores the gendered and embodied aspects of constructing a multiracial and multiethnic identity in a monoracially oriented society through in-depth semi-structured interviews with 11 self-identified multiracial and/or multiethnic women in various geographic locations across the United States. It describes, from a critical race and critical feminist approach, how multiracial women conceive of and navigate their physical bodies. This project explores the following specific questions: How do multiracially-identified women make sense of their identity in light of their mixed background? What role does skin color and the body play for the multiracial person themselves and the onlooker? and What do multiracial women have to say about the social construction of race in the United States?

Many recent studies of mixed-race identity have aimed to answer these questions paying particular attention to groups of individuals outside the Black/white identity (Bettez 2010; Jackson 2012; Jimenez 2004; Moreman 2011). This is crucial because there has been an increase in population in the United States in more recent decades of other groups of people such as those from Asian countries and Latin America due to larger global economic forces and consequent immigration. My research builds upon these works by adding to the body of literature on individuals of Latina/white backgrounds and Asian/white backgrounds.

The article is organized first through situating the research questions in relevant interdisciplinary literature on multiraciality, resistance, race and gendered embodiment, followed by a discussion of research design and description of participants. Then, data from the interviews along with analysis related to the gendered and embodied process of multiracial identity development is presented.

*1.1. Multiraciality, Monoraciality, and the Body*

The United States has historically been a monoracially-oriented society which means that racial identity is treated as a singular identity category. Individuals only possess one racial identity. Historically, individuals from mixed backgrounds had no place in the dominant racial framework, or rather were forced into monoracial categories of identification through a process of hypodescent. Hypodescent was established early on in the colonial period to preserve white racial purity and privilege. Hypodescent is a general rule applied to groups of multiracial people. It is important to note that there has been more flexibility in identification for multiracial people other than biracial Blacks/whites, particularly those of one white parent and one parent of color. As Daniel et al. (2014) said, "monoracial claims originating in rules of hypodescent are the basis for normative patterns of identification in the United States". Monoraciality forms the racial framework of this project. Highlighting the dominance of monoraciality in the United States draws attention to how multiraciality poses a threat to the dominant racial paradigm.

Relevant to this research and the increasing population in the United States, a mixed Latino identity is linked to the physical body particularly through markers such as skin color. Those from mixed Latino and white backgrounds are often mistaken for white. A focus on mixed Latino identity can also be conceptualized as mestizaje. Mestizaje is a specific term applicable to people of Latin America and the American Southwest. This term also acknowledges the conquest and colonization that led to mestizaje. Sociologist and Chicana studies scholar Turner (2014) wrote, "25% of Mexican Americans marry someone of a different race/ethnicity. This percentage highlights the significance of examining the offspring of these marriages as they will form a large part of the population in the future" (p. 133). Anzaldúa (1987) addressed the issue of mestizaje and mixed race. Anzaldúa called for a positive reclamation of indigenous identity. Like many, she walked her life in the borderlands, a neither "here nor there" space similar to being "betwixt and between" (Daniel 2001).

Much of the discussion and existing literature on multiraciality and the body focuses on skin color and the representation of the multiracial body (Haritaworn 2009; Williams 1995; Young 1995). Representation of mixed-race bodies is present in a variety of work on multiraciality (Haritaworn 2007, 2009; Young 1995). Haritaworn (2007, 2009), a qualitative multiracial scholar, discussed the multiracial body, focusing on mixed women of Thai and non-Thai heritage in Europe. Haritaworn called for a critical examination of the mixed-race body that looks at how even positive exoticization can be harmful and oppressive. In addition, Sanchez (2004) argued that "the multiracial body has been appropriated for use as a symbol of multi-ethnic America, often representing the nation's hope for the future and its potential for overcoming racial strife . . . the multiracial body appeared in the 1990s as an angelic savior for our age" (p. 277).

Thompson (2009) analysis of biological racism included the role of the physical body and appearance, "Racial aesthetics is, above all, a mode of observing the body . . . the emphasis on aesthetics and appearance within biological racialism negates the existence of mixed-race individuals . . . Power is instilled in the materiality of bodies" (p. 7). She stated that the existence and political consciousness of multiracial people threatens the established racial order of white supremacy as multiracial people have no place within the existing racial structure. Therefore, due to the way race is inscribed upon the body, multiracial individuals are a powerful threat. More research is needed on the link between multiraciality and the physical body. Multiraciality provides a unique site to examine the biological imperative of race considering how multiraciality's connection to the physical body does not fit into the dominant racial framework.

Popular perspectives on multiraciality as a way to overcome racism in the United States do not challenge the racial hierarchy, racial essentialism, or white supremacy. Rather than challenging the core of the unequal system of race in the United States through a critical examination of multiraciality and monoraciality, these concepts return to race as biological and the importance of offspring from interracial marriages. A new perspective of multiracial bodies is needed to deconstruct monoraciality and its hegemonic presence, despite the increased number of mixed-race people in the United States.

## 1.2. Embodiment

This research explores multiracial bodily practices from an "embodied sociology" perspective (Bendelow and Williams 1998; Davis 1997). As Waskul and Vannini (2006) wrote in their introduction to *Body/Embodiment*, "the term 'embodiment' refers quite precisely to the process by which the object-body is actively experienced, produced, sustained, and/or transformed as a subject-body" (3). Embodiment is an active process through which individuals make meaning out of their bodies through interactions with others (Cooley 1902). Reflexive embodiment is the process through which individuals exercise their individual agency by manipulating, changing, and altering their physical bodies (Giddens 1991, 1992; Crossley 2006; Connell 1995). Campbell and Maynell (2009) wrote, "the human body is an agent, inevitably transforming through its actions both the world and itself" (1). However, reflexive embodiment can also occur as a result of cultural mandates in the example of women dieting for weight loss (Bartky 1990; Stinson 2003). Self-policing as a result of external panoptic mechanisms might be how Foucault (1979) would conceptualize reflexive embodiment in this instance.

The material form of the body is what allows individuals to be active agents. Warin, a scholar of Health Sciences, (Warin 2014) argued for the importance of taking both social constructivist accounts of the body alongside materiality of the body. Warin defined material feminism as "attending to the ways in which bodies interact with and are radically open to other bodies, different spaces, histories, technologies and environments . . . It argues for a redefinition of how we come to understand embodied relationships between the natural and social" (p. 52). An exploration of embodiment from both a material and social perspective is necessary for research on multiraciality especially as it relates to gender and sexuality.

The body intersects with other forms of difference such as race and gender (Butler 1989; Bartkowski 2005). In terms of race and embodiment, for example, historical representations of Black women are strongly linked to gluttony and sexuality (Beauboeuf-Lafontant 2003; Strings 2015; Collins 2000; Townsend Gilkes 2001; Schott 2016) and common historical stereotypes of Black women often revolve around the shape of the physical body. For example, the stereotype of the Black matriarch is often depicted as a large, effeminate and unattractive Black woman. This stereotype served the purpose of blaming Black women for the supposed decay of the Black family and the emasculation of their Black husbands and continues through popular tropes today (Collins 2005).

The pursuit of thinness is an embodied way to gain status and increase mobility. Thinness is connected to privilege and whiteness. For women of color, the achievement of thinness can be used as a step up in a society where they face both racism and sexism. Through the pursuit of thinness through exercise, Black women have historically attempted to distance themselves from common stereotypes and the legacies of slavery (Purkiss 2017). This study explores this phenomenon in mixed-race women of a variety of racial and ethnic backgrounds, expanding the state of the research on race and embodiment. This research also adds to the bulk of literature discussing the bodies of women of color from a variety of racial and ethnic backgrounds.

## 1.3. Resistance

The theme of racial resistance is a critical component of studies on race, multiraciality, and monoraciality (Desfor Edles 2003; Jackson 2012; Romo 2011; Spickard 2003). There are many ways that multiracial people resist the dominant racial order in their everyday lives. For example, Romo (2011) showed how her participants performed race and asserted a Blaxican identity as tactics of resistance.

In order to be seen as legitimately Black or Mexican by both outsiders including the Black and Mexican community, the respondents had to choose one of the sides, not both. The majority of Romo's participants rejected forced monoracial identification by embracing and asserting a mixed Blaxican identity.

Another way to resist is through revealing knowledge that challenges historical myths about racial mixing. Desfor Edles (2003) shed light on the ways the women and other participants in her study challenged myths about racial mixing in Hawai'i. The high number of multiracial intermarriages creates the illusion that Hawai'i is a paradise of diverse multi-racial relations. This myth obscures the history of colonization, imperialism, and oppression in Hawai'i, masking the currently existing racism. Revealing this history is a way of resisting the dominant racial framework of the United States. Spickard (2003) discussed resistance through multiracial studies and whiteness studies. On the one hand, these two fields of study can support colorblind racism—the belief that the United States is a postracial society no longer affected by racism– and white supremacy (Omi and Winant 2014). But they can also be used as sites of resistance, "I have written that multiracial people, by their very choice to assert a multiracial identity, are 'undermining the very basis of racism, its categories" (Spickard 2003) (p. 291). Therefore, similar to Romo, those asserting multiracial identities resist the dominant racial order.

Feminists have also emphasized the potential for bodies as sites of struggle and resistance. Campbell and Maynell (2009) wrote, "the human body is an agent, inevitably transforming through its actions both the world and itself" (p. 1). Feminist scholars including Bordo (1993); Bartky (1990); Frye (1983); Young (2005); Collins (2000); Lorde (1984, 1986); Nkweto Simmons (1999), and Williams (1991) challenged the dominant white feminist perspective on the body and "have called on feminists to see that the ability to ignore the body in theorizing positive agency rests on the ignorance and privilege of these bodies that have not been marked by modes of oppression other than gender" (p. 10).

## 2. Methods

For this research, 11 women were interviewed who self-identified as multiracial, multiethnic, and/or mixed race. In order to be included in this project, the participant must have identified as either multiracial and/or multiethnic. The ages of the women ranged from 20 to 35 years. The women were interviewed within this age range for two reasons. First, they were chosen to obtain an adult perspective on the participants' life in terms of race and ethnicity and it was assumed that women this age would have had some time to reflect on their racial and ethnic identities. Secondly, women above the age of 18 years were selected for ease in the Human Subjects process so as to not need parental permission. The participants lived in various locations across the country from Portland, Oregon to Boston, Massachusetts to North Carolina and California and were recruited through snow-ball sampling relying on acquaintance referral. I began with multiracial women I knew through friends and those I had met in Santa Barbara. I was put into contact with the women through email. I emailed the participants and either set up a phone or skype call (if they lived outside of Santa Barbara) or set up a meeting time and place (if they lived in Santa Barbara). Most of the women I contacted agreed to be interviewed. I told them that I was interested in how their experiences as multiracial and multiethnic women informed their identity development, particularly in relationship to food and their bodies. The study was approved by the Human Subjects board at UCSB and all participants signed informed consent forms. All names mentioned in this paper are pseudonyms in order to preserve anonymity. All interviews were completed over the summer of 2015 and the fall of 2015. Participants were asked a variety of questions relating to their family, life growing up, racial/ethnic identity formation, and their relationship with food and their bodies. The women came from a variety of racial and ethnic backgrounds. The majority came from middle class families and attended secondary educational institutions (see Appendix A Table A1 for details on participants).

I pursued this research as a scholar engaging in research about a group in which I considered myself to be a member. My status as an insider was central to this research. My methodology and theoretical perspective followed to conduct this research stems from feminist methodology and critical ethnography. Critical ethnography is a specific approach to qualitative methodology and theory extending beneath surface level interaction, aiming to reveal underlying power structures steeped in historical and social constructions affecting the lived experiences of individuals (Harvey 1990). Critical ethnography allows the researcher to bring those at the margins back in to the center. Critical ethnography connects lived experience to history and socially created structures of inequality (Bhavnani and Talcott 2011). My research methodology questioned dominant forms of knowledge production and what counts as "real" knowledge in what I call feminist critical mixed-race studies. The purpose of my research was to give voice to these lived experiences while exploring and centering the role of the body in mixed race identity. Central to my perspective is the concept of intersectionality, the idea that categories of identity such as race, class gender, and sexuality are overlapping and intertwined (Crenshaw 1989). My status as an insider led to many advantages while conducting my study but I must also acknowledge my limitations due to my positionality. My positionality and reflexivity are vital aspects of this project. My role may cause some to question the validity, objectivity, and generalizability of this study. I make no claim to any of these positions (Collins 1986; Smith 1999).

I conducted qualitative interviews using a semi-structured interview guide with the research participants. My feminist critical mixed-race studies perspective led me to the method of qualitative interviewing as a way of featuring the voices of the participants. I used a semi-structured interview guide but let the participant lead the discussion, allowing the interview to happen on the terms of the participants. The interviews lasted anywhere from 45 min to two and a half hours. I digitally recorded the interviews using a program on my personal computer. Four of the interviews were performed over the phone or skype. The other seven interviews were conducted in person at a neutral location such as a café.

I transcribed and coded each interview, then printed out each transcription in order to code by hand. In my analysis, I employed an approach that combined sensitizing concepts (Charmaz 2003; Bowen 2006; Barrie et al. 2019) with grounded theory. Theoretical concepts were used initially to guide the identification of themes relevant to the theoretical framework I used to approach the interviews. Then grounded theory aided in revealing of further themes that emerged from the data that may have been outside the theoretical framework. A grounded theory approach lets the data speak for itself and allows themes to emerge based upon the dominant concepts in the data (Glaser and Strauss 1967; Lofland et al. 2007). Grounded theory is important in that it allows the participants in the study to speak to what is most important to them rather than only responding in a way that confirms the researcher's assumptions and interview guide. Qualitative interviewing with a semi-structured interview guide allowed for me to engage in this approach (Moreman 2011). I asked participants questions based upon my interview guide (informed by my theoretical framework) but also allowed them to take the conversation in whatever direction felt pertinent to them. I began with general coding by identifying themes common among the interviews. I then engaged in focused coding and narrowed down the themes that were important and relevant themes for the majority of participants. The codes I developed included resistance, knowledge, family, community, body, and whiteness. These focused codes allowed me to focus on the themes present in my analysis. It is important to note that these data and my analysis do not represent all multiracial people but rather, are specifically representative of the individuals within this project.

The number of interviews gathered for this research was limited, as only 11 multiracial and multiethnic women were interviewed. Therefore, the results of this project and data collection cannot be generalized to the larger multiracial female population. My findings are a reflection and a symbol of common experiences faced by multiracial people found in multiracial literature and based on 11 in-depth interviews. Another methodological limitation of this research is the lack of comparison

group. Results may have varied and provided richer data if a comparison group of multiracial men or monoracial individuals was included.

## 3. Results and Discussion

Three main themes emerged from the data. The first theme concerns how participants perceive their own racial and ethnic identities. The second theme concerns how participants understood their identities in relation to how others viewed them. And the third theme involves the role of the physical of body, which informs both their understandings of themselves and others' understandings of racial, ethnic, and gendered categorizations.

In this paper, I put forth the idea that micro-level exchanges among the participants and outsiders are potentially powerful instruments of resisting monoracial classifications. A variety of factors contribute to this resistance including the variables age, gender, sexuality, etc., in addition to race and ethnicity. It is not singularly race and ethnicity that determine the experiences of these participants but rather the intersections of these complicated structures. Therefore, it is the combination of being multiracial or multiethnic middle-class heterosexual woman in California (for example) that leads to the construction of these unique experiences and paths of resistance.

### 3.1. Their Own Ethnic and Racial Identities

At the time of our interviews, the majority of participants identified with a liminal or marginal identity (Daniel 1992; Park 1928; Thompson 2009). That is, in their own ways, they embraced a multiracial, in-between identity, which is a "both/neither" rather than an "either/or" monoracial identity. Many participants used various terms such as "multiracial", "mixed", and "in-between". Some purposefully and clearly engaged with these terms in order to outright challenge racial norms.

Although at the time of the research, most participants self-identified as multiracial in one way or another, their identities as multiracial fluctuated. For the participants, self-identification varied over time and was often flexible and context specific (Poston 1990). Jackson (2012) described shifting racial/ethnic expressions for the multiracial participants of her qualitative study. She found that many of the participants modified their racial/ethnic expression based upon the race/ethnicity of the people in their surrounding environment while some grew confident in their multiracial identity as time passed. My participants also demonstrated fluid identifications and expressions of their racial identity.

Ana grew up in the Bay Area of California with her Hawaiian/white mother and East Indian father. Her parents met in college when her father came from India to study. When asked, Ana strongly asserted her identity to me, "I identify as Asian Indian, white, and Hawaiian . . . Yes absolutely, I identify as multiracial". Ana clearly and confidently embraces a multiracial identity both to me in our interview and in her everyday life with those asking about her racial and ethnic identity.

Sanya, Ana's sister, whom I met through Ana rejected both a specific racial identification and a multiracial identity consisting of a combination of separate races/ethnicities. She explained to me when talking about the U.S. census, "it forces you to conform to one race. Even if someone is multiracial, they will not have the same experiences as you. I understand the census is trying to collect information, but it doesn't make any sense to me because you can't compare. They are defeating the purpose by lumping people into a category". Although the census is meant to put people into categories based upon similarities, Sanya is saying that putting all multiracial people into one category does not make sense because there is such a variety of races, ethnicities, and experiences among the mixed-race population. The census, a population-wide example of how race is conceptualized in the United States, as of 2000, has allowed respondents to check more than one box for race. However, there is no multiracial box to check. However, as Sanya points out, even a multiracial box lumping all multiracial people in to one category would not demonstrate the variety of multiracial people.

Sisters Sanya and Ana, in their self-identifications, question monoracial norms and classifications. Sanya questions the racial order in the United States as exemplified by the census. Sanya takes on an extraracial identity (Renn 2005) meaning she does not identify herself using any formal census

categories. Ana takes on a multiracial identity identifying herself as multiracial (Renn 2005). She rejects how she is "supposed to choose" to identify and embraces a liminal personhood. Checking separate boxes on a census reifies racial distinction, which defeats the purpose of identifying as multiracial. As Daniel et al. (2014) said, "Even the current formula . . . which allows individuals to check more than one box . . . puts forth the notion that multiracial-identified individuals primarily should view themselves as parts of various or multiple monoracial communities rather than also as constituents of a multiracial collective subjectivity" (Daniel et al. 2014). Sanya questions these issues and resists classification as designated by the census and U.S. as a whole.

Isabel, a woman of Japanese and European descent, grew up near Seattle, Washington in a relatively racially and ethnically diverse community. She spoke to me about how she checks boxes on the census and other demographic forms, "If you can pick more than one, I do Asian and white. If there aren't any other choices, I do other. I can't decide between Asian and white, so I choose other". Isabel's choice is an example of how someone begins to challenge the monoracial imperative and confront monoracial categorization. She has the opportunity to check white yet chooses not to do so. For Latinos in the United States who are not seen by the census as a racial group, checking the other box can be seen as a sign of resistance (Daniel 2001).

Myra also resisted classification but in a different way than any of the other participants. She said that she would mark Caucasian (white) on a survey or census,

> I guess if they saw me, they would wonder about that because I have darker skin. No one has questioned it or said anything about it. What would I mark? I have a German passport and my dad is as white as it gets. My mom's darker genes are dominant in the next generation: my sisters and I have color and dark eyes and hair, but I didn't feel that different. Only some people made me feel different.

Myra acknowledges her difference from other white people based upon skin color at the same time, she is aware of the whiteness of her German father. From Myra's perspective, her choice to mark Caucasian or white on surveys is a reasonable response to a racial classification system with no place for multiracial people.

I interviewed another set of sisters, Phoebe and Carson. They grew up on the West Coast living between their white mother and Latino father. Their stories are examples of how racial and ethnic identity change over time as well as how the physical body figures into gendered multiracial identification. Phoebe described to me how she believes she is both Latina and white but how other people don't see it this way, "It's like when I say I'm white people are like, 'oh but your last name is Garcia and you don't really look white' and when I say I'm Latina people say, 'oh you're white.' People don't think you can be both". This is interesting in light of the U.S. census and other forms that require respondents to declare race and ethnicity because on the census one can be both white and "Hispanic". However, as Phoebe shows, in her everyday life, being Latina and white are seen as mutually exclusive, especially when one has lighter skin color and can pass as white. She also said, "I've been more consciously identifying as Latina in the last few years because I've gained more of an understanding of multiculturality. Just because I'm white doesn't mean I'm not Latina". Phoebe points out common societal monoracial perceptions. Yet she clearly states multiple times that she believes she can be both Latina and white. Phoebe touches upon an important key to racial identity resistance in these participants' lives.

Carson, who can easily pass as racially white, also sees herself through a multiracial lens. For example, Carson said to me, "Yesterday at work someone asked me my last name. It was good. We led in to a very good conversation about labeling someone as Hispanic or Latino . . . ". I then asked her if during this conversation she was choosing to identify as both white and Latina and she said, "when I was talking to her I was identifying as both, but I think she only asked me that question because my last name is Garcia". Although at times she is perceived as only white, she still chooses to embrace a multiracial identity. This exemplifies how racial identity may vary situationally (Renn 2005).

Their identities and expressions challenged outsiders' notions of race and ethnicity, which in turn affected how the participants viewed themselves (Cooley 1902). A catalyst for discussion of race in the lives of sisters Phoebe and Carson was their last name, Garcia, a common Latino surname.

Carson's and Phoebe's identities can be viewed through a lens of multiracial studies and new Mestizaje studies as they are both multiracial and Mestiza. Turner (2014) outlined the similarities and differences, stating that one significant similarity is that both perspectives challenge inegalitarian concepts of race and racial mixing. Carson's last name not only led to a discussion about race and ethnicity but also was a venue for her to assert her white and Latina identity. Because she can pass as white, her claim to a multiracial identity and resulting educational conversation with a coworker are mechanisms for questioning monoraciality. For example, her coworker had assumed based upon her appearance that she was white, "He said I don't like curry. I've never met a white girl who doesn't like curry (referring to her). And I was like excuse me, I'm not white, but I look white to you". In this instance, she confronts this person's perceptions about race/ethnicity. Carson also challenges people's ideas of race and ethnicity through language, "I show up to job interviews and people ask me if I speak Spanish and I say no, even in New Mexico around white people they didn't get that I didn't speak Spanish". People with Spanish surnames are expected to speak Spanish. Carson defies assumptions of essentialist categories of race and ethnicity. In this paper I put forth the idea that micro-level exchanges among the participants and outsiders are potentially powerful instruments of resisting monoracial classifications.

Age was another variable that played into racial and ethnic identification over time. As they grew older, a few of the women gained a larger and more nuanced understanding of race. This knowledge created space for them to embrace multiple identities. For example, through education about race and culture, Phoebe chose to embrace her father's race and ethnicity. When I asked Isabel if she felt connected to her Japanese side she said, "I like to think so. I like the Japanese culture, and I studied the language and gained some of that culture back by learning Japanese. But, when I visit Japan I don't feel like I really belong. "As Carson and Isabel gained knowledge of their respective cultures, they felt more comfortable identifying with said culture and in Phoebe's case, the acquired knowledge not only allowed her to negotiate her identity but also demonstrated her deep understanding of the roots of race and privilege in the United States. Daniel et al. (2014) asserted, "multiracial identity formations interrogate monoracial norms supporting notions of white racial purity as well as European Americans' investment in whiteness and its attendant privileges" (p. 13). This section addresses how participants chose to self-identify racially and ethnically in a society that challenges their mixed identity. Many of these women engaged with their identities and in doing so questioned dominant monoracial norms of classification.

### 3.2. Others' Perceptions of Their Racial and Ethnic Identities

The women I interviewed lived as insiders and outsiders as they navigated between different social environments throughout their lives. These individuals are capable of revealing embedded and socially constructed notions about the complicated intersection of race, ethnicity, and gender in the United States. Race is socially constructed through relationships and interactions between outsiders and insiders. Racial and ethnic insiders (the multiracial women in my study) construct their identities in contexts affected by outsiders (those who do not self-identify as part of the same gendered and raced category). The previous section shows how the agency of the women strongly shapes their chosen self-identities. This section demonstrates how a gendered racial and ethnic identity is constructed in tandem with the perceptions and actions of outsiders through shame and embarrassment, and racism.

Navigating two worlds or existing on the border (Anzaldúa 1987) as a mixed-race woman can lead the individual to knowledge acquisition, giving her tools to question and possibly resist the dominant racial order. This position is a challenging place to exist particularly in the United States, yet it can provide opportunities for marginal individuals to gain deep insight into the complexity of intergroup relations. Although it is possible for many types of people to resist a racist society and

engage in antiracism, these women's positions as witness to outsiders' perceptions provide them with a unique understanding of race and racism. Two subthemes emerged within the larger theme of others' perceptions of the participants' racial and ethnic identities. The first is the shame and embarrassment participants felt in relation to these perceptions and the second is the racism they experienced at the hands of others. Below I review results with respect to the subthemes of shame and embarrassment and racism.

### 3.3. Shame and Embarrassment

The way others perceive and act towards multiracial people is an essential element in determining how multiracial participants feel about themselves. Therefore, the role of outsiders plays a pivotal role in the development of a multiracial identity. Many participants explain feeling ashamed and embarrassed by their racial/ethnic background when they were younger. For example, Carson told me, "not wanting people to know my last name, embarrassed (ME?) and I thought people would think I was Mexican". And Phoebe, "when someone reads roll call and I get butterflies in my stomach, I feel like people are looking at me like wtf: that white girl has the last name Garcia . . . Whenever I'd hear my last name, I'd be like ughhhhhh". Being Mexican was a source of shame and imbued with a negative connotation in the white dominant environment where Phoebe and Carson grew up. Because they can physically pass as European American, they rejected a part of themselves that could be associated with being Mexican and instead embraced whiteness. This was made possible because the dominant and most salient race in their immediate environment was white.

When Carson was in high school, a Latino boy found out her last name was Garcia and asked her, surprised, if she was Mexican. Carson responded, offended, that the name was Spanish, not Mexican. Particularly within New Mexico, one can claim a higher status as "Spanish" or Hispanic rather than Mexican. She did not want the boy to think she was Mexican and thus of lower status than she believed herself to be. This suppression of multiraciality by the society around them, a parallel to what Collins (1986) discussed as suppression of Black women and Black feminist thought, for some, is a catalyst for resistance; to fight against what once oppressed them. Awareness and a critical mindset towards their past experiences led them to embrace a multiracial identity and put them on a path to resisting monoracial classification.

Shame and embarrassment led Phoebe and Carson to reject the Latino part of their identities at a young age. However, both sisters eventually grew to accept and maintain cultural ties to their ethnicity, which increased as they became older. Carson provides an apt example. She brought up colonization when discussing how she sees the difference between Hispanic and Latino,

> To me Hispanic it means of Spanish origin so it's like recognizing a conquered colonial past which to me, most people, if they are closer to an indigenous root, are less likely to accept that term. It seems the whiter people are, the more they want to associate with a Spanish lineage than an indigenous lineage. Latino is the safer term to use. I would never call someone Hispanic unless they identify as Hispanic.

She distinguished between Hispanic and Latino based upon her knowledge of a history of colonization. Her knowledge is based upon how she sees certain groups of people's reactions to claims of being Hispanic versus being Latino. She equates identifying as Hispanic as more Spanish than indigenous. Her family is from New Mexico, which has a history of native populations choosing to identify as Hispanic or Spanish rather than Mexican or Indigenous in order to distance themselves from an oppressive past and approximate whiteness (Zavella 1993). She then goes on to say,

> I would never call myself Mexican. I say who is to say what Mexican is, when all of your ancestors lived in what was used to be Mexico. It's so complicated when you get indigenous blood in there because we got that through rape or persecution, or something fucked up. My ancestors' bones are in the mountains of New Mexico including my father's.

Her family and ancestors are native to this part of the United States, which used to be part of Mexico. This interplay combining her personal choice of identification and awareness of the role of outsiders, is a powerful method of resistance on a micro-level. Carson connects hegemonic racial structures imposed by outsiders to her daily life and how she chooses to identify.

### 3.4. Racism

Due to their liminal positions, many multiracial people witness countless acts of racism from outsiders. Witnessing racism and naming it led women to question the dominant racial order. Collins (1986) discussed Black women existing in a traditionally white and male space. Similarly, mixed race women exist in a world geared towards monoracial identification. An example of this is Marie who lived between two worlds growing up in Albuquerque, New Mexico. She talks about her dad's more Spanish side versus her mom's more Mexican side. Her father remarried a white woman from Pennsylvania and moved their family to North Carolina. Marie learned to navigate a variety of different spaces throughout her life. She described how New Mexico is different from North Carolina in terms of racism, "People are proud of it (the confederate flag) and say the history of the South is rich and represents history. And this kid I knew wanted to put it up (hang it up outside his house) and I asked him about the history he was talking about and he couldn't tell me about this history he was passionate about". Marie also acknowledged racism between those who identify as Spanish and those who identify as Mexican in New Mexico (Zavella 1993), "That side is more Spanish than Mexican which is different too. Nana always talks about how her dad and mom wouldn't let her talk to this Mexican boy she had a crush on". Marie's experiences and family life led her to gain a unique understanding and perspective on race. She was able to question her friends in North Carolina about their pride in the South's history and thus to question racist justifications.

When we were discussing her family's class background, Katherine brought up the topic of racism between darker skinned people and her lighter skinned Mexican family, "Proud Mexican old ladies. Very light skinned, blue eyes, tall. My grandpa was very short with brown eyes, but light skin and hair and it was probably an arranged marriage, but she wouldn't want to admit that. There's a lot of racism, with darker skin Mexicans, Blacks, doesn't matter, it's super racist". Within Mexico, lighter skinned Mexicans often have more privileges and higher status in society due to their closer phenotypic approximation to Spanish ancestry. Coming to the United States, her family may feel like they need to compete with other racial and ethnic minorities to boost their status.

Myra told me, "I used to reject the sari. My mom represented what was not loved and respected, so I did not want that. I did get the idea that white people are better. My mom was not that respected by the community. They didn't fully take her seriously. They expected her to be practical. I didn't want to identify with that". Myra is clearly aware of racial dynamics present growing up around mostly white people. Because the community did not respect her mother, Myra also did not respect her mother. It is possible that Myra may feel differently about the situation if instead, her father was Indian and mother white. How having a parent of a specific racial and ethnic identity affects a child's racial development is an area to be further explored in forthcoming research.

Witnessing racism by outsiders led many to question and take note of the role of race in society. This awareness, gained over a lifetime, aids the participants in developing a resistant consciousness.

### 3.5. The Physical Body

The thread underlying the perceptions of self and the role of outsiders is the physical body and embodiment of the participants. Participants engaged their bodies and knowledge of bodies' connection to ethnic/cultural histories as a way of developing a gendered multiracial identity and resisting monoracial identity imperatives. The body as one of the main symbols of race in our society plays a significant role in the lives of these multiracial women. For these women, themes of covering and passing, and confidence emerged. As a feminist critical mixed-race scholar, the importance of the role of the physical body in the lives of these participants is key to my research. Feminism's

continued emphasis on women and gender's connection to the physical body underlies this study (Butler 1993; Wolf 1990). I demonstrate how the body's role is tied to the intersection of race, class, gender, and sexuality and how the body is strongly connected to multiracial participants' awareness of the dominant racial order in the United States.

The participants' paths to self-identification as multiracial individuals was facilitated by their own and other's perceptions of their physical body. Many of the women's chosen racial and ethnic self-identification emerged through a lifelong process of negotiating between societal expectations of race and their internal feelings, family dynamics, and racial and ethnic background. In some cases, one factor in how participants chose and currently choose to racially and ethnically self-identify is based upon the way their physical bodies are perceived by others. Carson and Phoebe can both easily pass as white. Earlier on in their lives they identified as white, not as multiracial. However, as they grew older and acquired knowledge about how race operates in the United States, both due to their subject position and their education, they began to question how their physical bodies were viewed by others, creating awareness of race and its connection to their experiences as multiracial women.

Two subthemes emerged within the larger finding of the importance of the physical body in the lives and identities of these multiracial subjects. The subthemes consist of covering and passing, and confidence and acceptance.

*3.6. Covering and Passing*

Throughout their lives, a few of the participants completely broke with half of their background and passed as white (the majority of participants are half white, see Appendix A, Table A1 for more information on participants). Passing is on the extreme end of covering (Daniel 2001). Covering is defined as disguising one's physical body in an attempt to distance oneself from a stigmatized identity (in this case, the intersection of being a woman and being mixed race) (Yoshino 2006b). The majority of the participants engaged only with various methods of covering (Yoshino 2006a). I highlight the physical aspects of passing and covering to show how the women in this study presented their physical bodies in order to cover.

A powerful part of covering is how one fits in physically to the dominant phenotypical model, "Overall, skin color along with other phenotypical features such as hair texture, eye color, as well as nose and lip shape working in combination with attitudinal, behavioral, and socioeconomic attributes has increased as a form of 'racial capital'" (Daniel et al. 2014, p. 24). The physical body is inexorably linked to race and gender (Butler 1989, 1993; Omi and Winant 2014). Historically, light skin has been the prerequisite for passing. I argue that based upon these participants' stories, passing and covering are heightened by other attempts at body modification. These types of modifications include the lightening of skin and avoidance of tanning to bleaching of eyebrows.

For the participants, the path towards developing an embodied multiracial consciousness was fraught with contradictions that in some cases also upheld the monoracial hierarchy. The women's relationships with their bodies demonstrate how they do not need to be free of patriarchal constraints on the female body to be resistant to a monoracial society. Bunsell (2013) in her ethnography on female bodybuilding in the UK wrote, "I find myself becoming increasingly skeptical of second-wave feminist claims that there is a 'female body' which can be 'reclaimed' from 'patriarchal' society ... there can never be a resistant 'female body outside of discourse, or a resistant body that can stand as a simple exception to forces of normalization or domination'". Like Bunsell, I find that the multiracial women participants in my research are complex subjects whose bodies play an often-contradictory role. However, these contradictions do not prevent them from questioning dominant structures.

Phoebe is an example of a participant engaging with her body from the intersection of gender, sexuality, and race. She recounted a memory of the past, "(in) junior high, I was obsessed with being as white as possible. I would wear long sleeves and pants in the summer so I wouldn't get tan, bleached my hair, bought skin-bleaching lotion and I'm not even that brown! I didn't want anyone to think I was close to Mexican or anything". I asked her why and she said, "I was embarrassed. All the kids

who were Latino—most of them Mexican—they had such a bad reputation as being bad kids and (having) bad families and stuff".

Then as she grew older, "In high school I was like, 'oh fuck that shit I'm going to wear dirty overalls every day and chop my hair off and wax my eyebrows off.' And as much as that was rebelling from those societal norms it was within those norms". She altered her body in a different way—by trying to look "different" than she did before and by appearing "alternative" and androgynous. And now, after college, she embraces her body and is now mostly comfortable with herself (Hooks 2003; Walker 2001).

Phoebe's presentation of her body is a powerful statement in a monoracial society. This is demonstrated through her path to embracing a multiracial identity. She is aware of how passing and covering in the past were reactions to not fitting in to society. Knowledge gained through her education and subject position led her to see this and then to embrace a multiracial identity. Today Phoebe embraces a fuller body shape. She is also aware of how others view her body. When she says "my body is Latino" she means that the way her body is perceived by outsiders is as a Latina body; she has wide hips, a big butt, and breasts—all features stereotypically associated with the bodies of women of color and in this case Latinas specifically. Phoebe's nuanced awareness of her past relationship with her body and her racial/ethnic identity in addition to how she is perceived physically today demonstrate monoracial knowledge. Both Phoebe's and her sister Carson's awareness of societal ideals and normative about women's bodies allows them to begin to resist normative monoracial expectations and make decisions about their own physical bodies in today's society.

Myra also felt self-conscious about her physical body and attempted to alter herself growing up as a mixed-race woman in Germany. She was often the only person of color at her school and the people who looked like her were maids and of a lower-class status. She told me, "I had horrible moments in my life when I wanted all my facial hair to be blonde, my eyebrows too much work to tweeze, I lasered it once. People said I had a moustache. I have the Frida Kahlo look. It felt like self-hate about it, I wanted to be like the people that are popular". Although less extreme than complete passing, Myra, like Phoebe tried to change her physical body to look more like the blonde European ideal. As mixed-race women, some of the participants recognize how they engaged in covering to alter their physical selves in order to fit into both the standard beauty ideal and negotiate a multiracial identity.

### 3.7. Confidence and Acceptance

Ana and Sanya both resist physical monoracial norms. They do so through the need to not define themselves physically as either Indian or Hawaiian. They are aware of how others perceive them and their family—as racially ambiguous. Not fitting into a dominant and expected phenotype for Indian or Hawaiian people compounds Ana and Sanya's multiracial self-identification and conviction in their in-between identities. As Desfor Edles (2003) discussed in her piece "'Race', 'Ethnicity,' and 'Culture,' in Hawai'i, The Myth of the 'Model Minority'", there is a popular conception of Hawai'i as a mixed-race Utopia. Ana and Sanya both have an awareness of history on both sides of their family and how this history has affected their family and themselves including dispelling the myth of Hawai'i as a racial utopia. This awareness allows them to make decisions about their physical bodies. The sisters consciously exist within a liminal physical space neither attempting to look phenotypically or stereotypically Indian or Hawaiian.

Marie told me, "I never felt negative about my body because my mom was super positive . . . I've never really felt unconfident . . . I've never been one to watch or count calories I just like enjoying food". Marie mentions her mother as a key figure in her acceptance of her body. Later in our conversation she also mentions the positive effect of her grandmother, who had a large hand in raising her. Scholars have long debated the role of mothers in the lives of their children (Freud) and while mothers do affect the way their children see themselves, as scholars like Thompson (1994) have pointed out, there are many other reasons such as racism, classism, and heterosexism that affect one's relationship with food and their body.

Carson also exemplifies the connection between physical appearance and multiracial identity. When it comes to her body she explains how she feels in an either/or monoracial framework, "If I was actually Mexican I could have big boobs and hips, and it would be good and sexy but if I'm white then I have to be skinny, very consistently that goes through my head". Keeping with cultural and social body ideals, she is "allowed" to have big hips and breasts if she is Mexican, but if she is white she must be skinny. She struggles between feeling the pressure to be skinny as a white person and the ability to embrace a curvier figure as Mexican. This struggle is strongly facilitated by her identity as a multiracial individual. Because she does not fit neatly in to specific phenotypical racial categories she is ambivalent about how her physical body fits in to the framework. In Carson's quote, she is playing off the stereotype that women of color are supposed to be heavier, have larger butts, hips, and breasts, while white women should be thin. But this limited and dichotomous view of ideal bodies does not leave room for many women in any category, above all not multiracial women. Although not explicitly stating this, Carson is pointing out that she does not know where in this dominant discourse about "ethnic" beauty standards she fits as a multiracial woman. Carson demonstrates both ambivalence and confusion towards how she is supposed to look while at the same time telling me that while it is a hard process, she tries to accept her body as it is now.

A powerful way of resisting society in general and society as monoracial was through women's acceptance and love of their bodies. Many of the participants embraced their physical bodies and accepted them as they are. This is the case of Marie with Katherine. I suggest here that one factor in their ability to accept their physical bodies is their approximation to ideal standards of beauty. Marie, who identifies as "Hispanic" may feel less pressure to be skinny because of society's expectations of the bodies of Latinas. For Katherine, who passes easily as white, her slender figure, which closely approximates the white slender societal ideal may allow her acceptance of her physical body.

## 4. Conclusions

Results reveal the variety of ways that multiracially identified women become conscious of the dominant monoracial order and begin to resist monoracism in their daily lives. I see this research as contributing to the study of multiracial populations in many ways. First, my study illuminates the lives of Latina/white and Asian/white and East Indian/white women, all understudied populations in literature on mixed race. My analysis of passing, covering, and the body document the spectrum along which people use their physical bodies to both resist and maintain the racial binary through awareness of how race is connected to gender and the body.

Possessing a marginal identity is a pathway to acquiring a resistant consciousness to monoraciality. This identity can be a catalyst for multiracial individuals, particularly women, to reveal embedded and socially constructed notions about race in the United States and across the world. Collins (1986) discusses Black women existing in a traditionally white and male space. Similarly, mixed race women exist in a world geared towards monoracial identification (Daniel et al. 2014; Jordan 2014). Negotiating between separate racial and ethnic identities, as multiracial women, often forces the individual to notice taken for granted societal ideas about race. Being forced to confront entrenched ideas of race is an avenue for multiracial individuals to see how race is socially constructed rather than biologically determined, and put them on the path to monoracial resistance. The women I interviewed are examples of the diverse ways a person can combat an oppressive monoracial society on an everyday basis through awareness of the connectedness of race, gender, and the physical body.

An area to be further explored in future research is the connection between thinness, status, gender, and multiracial identity. Previous research shows how slimness has been marked as a sign of status (Bordo 1993; Cheney 2011) and in order to obtain status, individuals attempt to achieve thinness. This study points to an important link between these factors which is worth probing through forthcoming research.

**Funding:** The research received no external funding.

**Conflicts of Interest:** The author declares no conflict of interest.

## Appendix A

**Table A1.** Participants, racial/ethnic self-identification, age, location.

| Name | Racial/Ethnic Self-Identification | Age | Location (at Time of Interview) | Class and Educational Status |
|---|---|---|---|---|
| Ana | Hawaiian/East Indian | 20 | Santa Barbara, CA | Upper middle, in college |
| Carson | Latina/White | 23 | Portland, OR | Middle, high school education |
| Celeste | Latina/White | 23 | Santa Barbara, CA | Middle, post grad education |
| Isabel | Japanese/White | 26 | Seattle, WA | Upper middle, college educated |
| Katherine | Latina/White | 25 | Santa Barbara, CA | Middle, college educated |
| Myra | East Indian/White | 34 | Santa Barbara, CA | Upper middle, college educated |
| Marie | Hispanic/Native American/White | 21 | Gastonia, NC | Middle, college educated |
| Phoebe | Latina/White | 22 | Boston, MA | Middle, college educated |
| Rebecca | Black/White | 24 | Santa Barbara, CA | Middle, post grad education |
| Sanya | Hawaiian/East Indian | 24 | Santa Barbara, CA | Upper middle, post grad education |
| Sasha | Latina/White | 22 | Santa Barbara, CA | Middle, college educated |

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
