# Peer review of "Embodied Resistance: Multiracial Identity, Gender, and the Body"

_socsci, doi:10.3390/socsci8080221_

Round 1

Reviewer 1 Report

The paper is much improved.  Having said that, Section 3.7  cannot be fixed without a comparison group and thus should be omitted.  In other words, without having a comparison group, it is unclear how almost all of the contents of this section is relevant to multiracial identity.  

Author Response

Thank you again Reviewer 1 for your comments. I have taken your suggestion and omitted the majority of section 3.7. I moved one piece of 3.7 into the last section on the physical body because I felt that it was significant, you can find this change on lines 672-687. I also included a few sentences in the conclusion about areas of further research including looking at how thinness is connected to gender and multiracial identity specifically. Thank you!

Reviewer 2 Report

I commend the authors on a thorough revision. I would raise two remaining concerns.

With the paper now featuring stronger theory (critical theories of embodiment and race), is it true that grounded theory was used to analyze the data? It might be more fair to say that the author used theoretical concepts to guide initial passes through the data and then used a grounded theory approach to identify themes that emerged from the data. See this article for an example of this two-step approach: https://doi.org/10.3390/socsci8030083. I am not asking the authors to misrepresent their analytical approach. I just think of grounded theory as not typically possible because researchers who know the literature and have a theory always bring assumptions to the data analysis phase. (No shame in that, of course. I find grounded theory of limited use.)

The references are not fully consistent. Some have quotation marks around titles, while others do not.

Well done.

Author Response

Thank you again Reviewer 2 for your comments and suggestions. I have taken your suggestions and incorporated a discussion of sensitizing concepts along with grounded theory in my methods section, including a citation for the Barrie et al.article. I have also edited and reformatted the works cited section so that all citations are consistent. Thank you!

Round 2

Reviewer 1 Report

I am glad you were receptive to the suggestions that I believe strengthen the paper. 

This manuscript is a resubmission of an earlier submission. The following is a list of the peer review reports and author responses from that submission.

Round 1

Reviewer 1 Report

This paper tackles an important topic but is overly ambitious in trying to present conclusions. The interviews touch on a surfeit of intervening variables beyond being multiracial, a severe limitation that is exacerbated by a small sample size.  The result is a paper that is descriptive but with an analysis that unfortunately does not allow us to attribute body negativity to being multiracial.

Questions about Methods:

How was the sample solicited: via email, text or a call?  How did the interviewer present herself in terms of her goals and how the data would be used/presented? What was the response rate?  Was this study approved by an IRB?  Was there an informed consent form?  Did the author use pseudonyms for the interviewees? 

How might the results have been different with men?  What is lost by not having a comparison group of either males or monoracial individuals (for at least some of the questions)?

While you discuss which parent is of a particular race regarding the surname at one point (e.g., the name called during attendance/roll), you do not delve into how having a mother versus a father of a particular race may affect identity. 

Literature

The authors need to document from the literature the perceived difference in the bodies of women of color.  Relying on just a few citations, e.g., Omi and Winant, is not sufficient.

Difficulty Drawing Conclusions from the Data

The main weakness of this paper is that without a comparison group of non-biracial individuals of color, we do not know how many of the findings can be attributed to multiracial identity versus just being non-white (or in some cases, just being female). 

In some cases, the author specifically mentions a particular interviewee’s commentary that attributes issues to being multiracial and in those instances, it is clearer. Nevertheless, with so few respondents, it is hard to see a pattern when a number of the issues could very well be part of being female and not necessarily related to race. We know, for example, that eating disorders are common, especially among white females, so it’s hard to be convinced by the limited interview data that issues broached are particular to the interviewees rather than reflective of life as a US female, as the author herself points out on line 495: “The pressure to be thin pervades American society.”

On lines 474-494, the author discusses issues of comfort with one’s body and brings in the Indian caste system.  This exemplifies how the author brings in too many extraneous variables that may play a role, but with such limited data, runs a real risk of conflating various interrelated factors mentioned by the interviewees.  For example, in lines 485-8, the author says:

Sanya connects issues of status and slimness to the caste status of her grandmother: “my grandma is very concerned with status when she married my grandfather it was pre-independent India, well, post actually, when they met. The caste system was in full force and they were different. I'm sure she's very aware of class and caste.”

The use of this quote suggests that status is partly determined by slimness, due to the influence of the caste system.  Yet there is no documentation of this, nor is it explained.  What is it about the caste system that aggravates pressure for women to be thin?  I could see if the caste-related issue were about social class and a family’s financial resources, but there is no explanation for why the caste system matters over and above the “the pressure to be thin [that] pervades American society.”

While the author does allude to the desire to be slim by citing “status through whiteness (String 2012)”, she does -not- include String in the citations and in any case, this point would need substantial corroboration; making that statement is not unlike saying the same thing about a high income (associated with whiteness but which is also desirable aside from such associations).  

The following paragraph about Myra is illustrative of the main weakness of the paper:

It tells us that Myra is:

1.     anorexic

2.      “is often mistaken for a variety of races and ethnicities.”

3.     Elicits a memorable reaction when dressed in visibly different clothes associated with her mother’s culture

4.     grew up in Germany, a country that has its own issues surrounding racial purity

5.     was the only person of color at her school

6.     felt that her looks associated her with being a maid and as low class

7.     was isolated and lonely

Paragraph about Myra, (lines 527+): “I did some Indian dance and wore my mom’s sari and I look very Indian when I dress Indian and I guess it’s a very superficial thing it’s just the way I look. I have people call me their Indian princess. They like the idea of it. People love India. If I do the makeup like an Indian actress, it’s very high status as an exotic. It is in our society. So much about image, the surface. Myra explains how her physical appearance can make her seem more authentically Indian to outside audiences. We also discussed what it was like for her to grow up as a mixed-race woman in Germany. She was often the only person of color at her school and the people who looked like her were maids and of a lower-class status. She felt isolated and lonely.”

Of the seven factors listed above, how are we as readers supposed to figure out the role of these interrelated variables in causing Myra to feel that she: “wanted to be like the people that are popular” (line 544).  Wanting to look like the popular crowd is not unique to multiracial persons, persons with body issues, or those who have lived outside the US.  All of these independently-important factors are jumbled together, without convincing me that being multiracial has an impact over and above what women who do not face all of these factors experience.  How they describe these sentiments, i.e., how they express them, may be unique to them is interesting, but does not provide sufficient insight on how perceptions of their bodies and “societal pressures to be thin” (line 548) are linked to the independent variables in this study. 

I wrote the above paragraphs before I read the author’s own assessment of Myra’s thoughts as “messy nuances” (line 548).  But even without including the paragraph about Myra, I am left feeling confused about what I can reasonably and confidently conclude about the topic based on the data presented.

In other words, I am not convinced of the validity of the author’s conclusions on lines 550-1: “Covering by pursuing thinness is a way to tone down the undesirability of a mixed race body.”

I would assert that the extent of female agonizing about thinness is simply way too prevalent to link body issues to these variables proposed by the author.  

In fact, the author brings home this very point, by adding (lines 569-570): “mothers do affect the way their children see themselves, as scholars like Becky Thompson (1994) points out, there are many other reasons such as racism, classism, and heterosexism that affect one’s relationship with food and their body. 

Thus, mothers and heterosexism have just been thrown into the mix, in addition to a wide interviewee age range of 20-35, spanning different stages of life in which concerns about one’s body often vary just due to getting older, or getting married, etc. Furthermore, having such a small sample size is incompatible with taking on the surfeit of variables encompassed by Latina/white and Asian/white and East Indian/white women---with only just a single person (left out of the categories above) who is from arguably the most stigmatized group (those who have African or African American roots—Rebecca, in this study)--making the results confusing and lacking in cogency.

In the Conclusion, the author states:

“Being forced to confront entrenched ideas of race is an avenue for multiracial individuals to see how race is socially constructed rather than biologically determined.”

Who is being “forced” to confront these ideas?  The interviewees?  The readers?  I also don’t see how airing their views shows that they understand how race is socially constructed. 

“And put them on the path to monoracial resistance. The women I interviewed are examples of the diverse ways a person can combat an oppressive monoracial society on an everyday basis through awareness of the connectedness of race, gender, and the physical body.”

I do not see how the interviewees are “combatting” societal oppression.  They are showing that they had issues with being multiracial but what strategies were employed to make them more at peace with these tensions?   On lines 563+, the author discusses Marie and how she found her mother and grandmother to be helpful in counteracting negative influences that might have otherwise caused her to feel negative about her body.  But we are not told of any details.

How do we know whether multiracial men suffer from the same kinds of issues?  It is unclear to what extent any of the findings are generalizable.

Grammatical Issues and Typos

I was unable to point out all of the corrections needed, but here is a sampling of 30+ errors:

Line 82: A new perspective of multiracial bodies is needed to deconstruct monoraciality and its hegemonic presence, despite THE increased number of mixed race people in the United States.

Line 92: considering that multiraciality and its connection to the physical body does not fit in to [INTO] the dominant racial framework.

Line 106: This sentence doesn’t read well: Migration into Hawai’i ever since the 1700s was diverse and large in number, intermarriage between these populations over time produced a multiracial Hawaiian population.

Line 129: And the third regards [INVOLVES] the role of the physical of body which informs both their understandings of themselves

Line 145: Ana grew up in the Bay area of California [capitalize Area] with their HER Hawaiian/white mother

Line 156: Although the census is meant to put people in to INTO categories

Line 158: The census, a population wide example   should be: population-wide

Lines 176-7: “If you can pick more than one, I do Asian and white. If there aren’t any other choices, I do other.  [COMMAs ADDED]

Line 186: I guess if they saw me, they wonder about that. COMMA ADDED

Line 213.  Avoid comma splices and change as follows: “Yesterday at work someone asked me my last name. It was good. We led in to a very good conversation about labeling someone as Hispanic or Latino . . .

Line 216: Although at times she is perceived as only white, she still chooses COMMA ADDED

Line 219: Carson’s and Phoebe’s identities. ADDED POSSESSIVE FOR CARSON

Lines 238-9. “I like to think so. [PERIOD HERE AFTER “SO” INSTEAD OF COMMA].  I like the Japanese culture, and I studied the language and gaining some of that culture back by learning Japanese and when I visit Japanese [IS THERE A WORD MISSING HERE?] I don’t feel like I really belong.”

Line 240: As these women gained knowledge of their respective cultures, [COMMA ADDED] they felt more comfortable

Line 243:  the word “say” sounds off.  Maybe try “assert”

Line 250: Because of this, these [COMMA ADDED]

Line 275: “not wanting people to know my last name, embarrassed [ME?]and I thought

Line 277: looking at me like wtf: that white girl [COLON ADDED]

Line 282: most salient race in their immediate environment was whiteness. white.

Line 305: It seems the whiter people are, the more  [COMMA ADDED]

Line 314: I would never call myself Mexican. I say who is [a period was substituted for a COMMA]

Line 317-8: I mean indigenous roots my ancestors bones are in the mountains of New Mexico, my father’s.  THIS is unclear.  It’s ok to use [brackets] to fill in words that were not said but that make the meaning clear.  You also need an apostrophe after the s on ancestors—and maybe a comma after the word roots (but I’m confused, so I’m not sure how to fix this sentence).

LINEs 329-334  This section/quote confusing.  Clarify or omit

Line 344-5: Nana always talks about how her dad and mom wouldn’t let her talk to this Mexican boy they had a crush on, always acknowledged that there was racism between Spanish and Mexican people.”

Line 348:  Her interactions with others provided the site for her to begin to resist racism.

Site?  I don’t understand.

Line 349: When we were discussing her family’s class background, [added missing comma]

Line 357: Myra told me, “I used to reject the sari, my mom represented

Comma splice above, fixed as follows:

Myra told me, “I used to reject the sari. My mom represented

Lines 359-60: I did get the idea that white people are better, my mom was not that respected by the community, they didn’t fully take her seriously. They expected her to be practical, and organized and looked down on that.

Should be:

I did get the idea that white people are better. My mom was not that respected by the community. They didn’t fully take her seriously. They expected her to be practical, and organized and looked down on that.

ALSO:  looked down on “that”.  What is “that”?

Line 401: phenotypical model,  Should be: phenotypical model:

Lines 425-6: Change to this: All the kids who were Latino--most of them Mexican--they had such a bad reputation as being bad kids and [having] bad families and stuff.

Line 471: The quote above  . . .   When I began to read that quote above, I was very confused.  It is only AFTER I read it that the author attributes it to Ana.

Run-on sentences:

Line 492: it affects me unconsciously on the surface I really don't care,

Lines 527-8: I look very Indian when I dress Indian and I guess it’s a very superficial thing it’s just the way I look.

Lines 573-4: This is the case of Marie and WITH Katherine. I suggest here that one factor in their ability to accept their physical bodies are IS their approximation to ideal standards of beauty.

Lines 609-10: I employed a grounded theory approach, letting the data speak for itself and allows ALLOWED themes to emerge

Reviewer 2 Report

This study examines the intersection of race, ethnicity, and embodiment using 11 interviews. It would be improved with attention to these issues. 

Please move the Methods section so it appears before the Results section. 

Provide more detail in the Methods section. The sample needs a clearer rationale with inclusion and exclusion criteria more clearly defined. Almost no attention is paid to data analysis. Were theoretical constructs used as sensitizing concepts to govern the analysis? The logic of coding and interpretation needs attention. 

There is a lot of great body theory (Michel Foucault, Kathy Davis) whose work could more strongly inform this study. The study would be enhanced with more attention to this. See Bartkowski’s article, Faithfully Embodied, for a review. 

Please address the limitations of the study (11 interviews is a small sample and ethnography is sometimes a richer medium for studying embodied practices). 

Good work with room room for improvement.